# Global Patient Involvement in Sarcoma Care—A Collaborative Initiative of the Connective Tissue Oncology Society (CTOS) & Sarcoma Patients EuroNet (SPAEN)

**DOI:** 10.3390/cancers14040854

**Published:** 2022-02-09

**Authors:** Bernd Kasper, Kathrin Schuster, Roger Wilson, Sorrel Bickley, Jean-Yves Blay, Denise Reinke, Markus Wartenberg, Rick Haas

**Affiliations:** 1Sarcoma Unit, Mannheim Cancer Center (MCC), Mannheim University Medical Center, University of Heidelberg, 68167 Mannheim, Germany; 2Sarcoma Patients EuroNet, SPAEN, 61200 Wölfersheim, Germany; kathrin.schuster@sarcoma-patients.eu (K.S.); roger.wilson@sarcoma-patients.eu (R.W.); markus.wartenberg@sarcoma-patients.eu (M.W.); 3Sarcoma UK, London N1 6AH, UK; sorrel.bickley@sarcoma.org.uk; 4Centre Leon Berard, 69008 Lyon, France; jean-yves.blay@lyon.unicancer.fr; 5Division of Hematology/Oncology, Department of Internal Medicine, University of Michigan, Ann Arbor, MI 48109, USA; dreinke@med.umich.edu; 6Department of Radiotherapy, The Netherlands Cancer Institute, 1066 CX Amsterdam, The Netherlands; r.haas@nki.nl

**Keywords:** Connective Tissue Oncology Society (CTOS), multi-disciplinary management, sarcomas, Sarcoma Patients EuroNet (SPAEN), patient advocacy, rare cancer

## Abstract

**Simple Summary:**

Patients and patient advocates from Sarcoma Patients EuroNet (SPAEN), a global network of national Sarcoma Patient Advocacy Groups, and medical experts from the scientifically driven Connective Tissue Oncology Society (CTOS) came together on 9 November 2021 at an official ancillary event to the CTOS 2021 Annual Meeting. At the event, representatives of CTOS and SPAEN jointly discussed gaps and challenges in global sarcoma care and management. This resulting position paper highlights the main findings and possible future steps.

**Abstract:**

Sarcomas are a grouping of rare cancers with a wide variety of histological types that are difficult to diagnose and treat. This leads to many varying challenges not only for sarcoma patients, but also for doctors, researchers, and caregivers. Patient advocacy groups have an important role to play in rare cancers such as sarcomas, especially in collaboration with experts and their medical societies. To this end, patients and patient advocates from Sarcoma Patients EuroNet (SPAEN), a global network of national Sarcoma Patient Advocacy Groups, and medical experts from the scientifically driven Connective Tissue Oncology Society (CTOS) came together on 9 November 2021 at an official ancillary event to the CTOS 2021 Annual Meeting. At the event, representatives of CTOS and SPAEN jointly discussed gaps and challenges in global sarcoma care and management. This resulting position paper highlights the main findings and possible future steps.

## 1. Background

Sarcomas are a rare group of cancers and can be difficult to diagnose and treat. This leads to many varying challenges for sarcoma patients compared to patients diagnosed with more prevalent cancers. Collaboration is key: partners with differing priorities and interests can benefit from finding common ground, sharing resources, and working together. Consequently, patient advocacy has an important role to play, especially in cooperation with experts and their medical societies. This is even more important in rare cancers such as sarcomas [1].

Sarcoma Patients EuroNet Association (SPAEN) is the international network of national Sarcoma, GIST, and Desmoid Patient Advocacy Groups (PAG) working to improve the treatment and care of sarcoma patients through information and support, and by increasing the visibility of sarcoma research and care with policymakers and the public. The objective of their work with experts and medical societies, such as the Connective Tissue Oncology Society (CTOS), is to broaden the scope and add to and complement specialist expertise with patient advocates’ experiences and perspectives, as well as to leverage patients as partners and to collaboratively improve care and management of sarcomas around the globe.

Roger Wilson (SPAEN Honorary President and co-founder, sarcoma survivor since 1999) outlined in an email exchange with Bernd Kasper (Medical Oncologist, SPAEN Board member, CTOS Board member) a vision: “I am a believer that we work best when we work together, and we have created great examples (Desmoid and leiomyosarcoma roundtables and papers) in the past. If we can turn those examples into a long-term CTOS workstream covering all sarcomas and reviewing that work on a regular basis, we can together build a platform that researchers and research funders worldwide will take note of and pharma will relate to it very happily. It will equip the patient advocacy community with an evidence-based underpinning for challenging the regulation and management of treatment and care for sarcoma everywhere. It will also bring the mainstream of CTOS into understanding what patients can offer in helping them be ambitious about the targets we all have for the future. A wonderful vision to have.”

On the basis of a Kickoff Meeting on 9 November 2021 as an official ancillary event to the CTOS 2021 Annual Meeting, sarcoma experts and/or representatives of CTOS and experienced patient advocates discussed gaps and challenges in global sarcoma care and management. This resulting position paper seeks to highlight the main findings of the event and presents possible future steps.

The management and care of sarcomas is complex and can differ vastly between healthcare systems and/or countries. For the purpose of this paper, the four most relevant areas have been identified and are discussed, followed by a brief summary of potential solutions and/or initial joint projects.

## 2. Discussion Areas

### 2.1 Optimizing Clinical Care in Sarcoma

#### Early and Accurate Diagnosis 

Sarcomas are a rare group of diseases manifesting in a multitude of different ways and affecting almost any part of the body. Even if suspected, it often requires an expert pathologist to confirm the diagnosis. If a proper pathway is followed, referral to an expert multidisciplinary team (MDT) or surgeon should be instigated. Tumor size is a known prognostic indicator and early diagnosis is linked to smaller tumor sizes or disease spread, suggesting a better prognosis for the patient [2]. However, the rarity of sarcomas means that the majority of primary care doctors will see few cases during a career and these cases will most likely be dissimilar. What is less readily recognized is that doctors in secondary care and local hospitals are also unfamiliar with sarcomas. This can lead to a delayed and incomplete or inaccurate diagnosis, inappropriate excision and treatment decisions, or no follow-up care. It is agreed that comprehensive literature and data relating to these scenarios is not readily available.

Anecdotal evidence cited includes France, which has a strong network of specialist sarcoma centers, NetSarc. It has 26 centers evenly spread across the nation. Thirty-seven percent of patients are referred to a NetSarc center prior to surgery, while 55% have surgery outside the network. It was shown that for surgery performed outside a reference center, the rate of R0 curative resection is 50% lower compared to patients operated on within a NetSarc center—while the rate of R2 resections with macroscopic residual tumor is more than doubled. The authors conclude that “surgical treatment in a reference center reduces the risk of relapse and death” [3]. We have no evidence to suggest that any country is exempt from this diagnostic problem. Even in a nation as well-resourced as the USA, the approach to diagnostics for sarcomas is described from within as “broken”.

If earlier diagnosis is important, accurate diagnosis and biopsy are equally critical. The relationship between pathology and treatment is often not well understood by patients. There is an assumption that science can be left to those who have been trained and they can be trusted to get it right. Sometimes it seems sarcomas are also a mystery to non-reference center pathologists who rarely see these types of cancers. Since the 1970s, and almost without exception, the reports of specialist pathologists suggest that approx. 30–40% [4] of initial diagnoses made outside a sarcoma expert network are incorrect. The impact of an incorrect histological diagnosis can be substantial, and thus of great consequence for the patient. Different sarcomas manifest and spread in different ways and this affects the approach to surgery. Tumor size may affect specific decisions regarding neoadjuvant therapy. Tumor subtype and grade may affect the approach to adjuvant therapy. Some tumors are more indolent than others, affecting follow-up decisions. There are many other treatment and survival facets affected by the pathology; hence, an accurate diagnosis is every bit as important as an early diagnosis.

### 2.2. Overcoming Healthcare System Challenges for the Management of Sarcomas

#### 2.2.1. Specialist Care of Sarcomas within a Multidisciplinary Team

The complexity and heterogeneity of sarcomas demand that patient care is best managed by a multidisciplinary team (MDT) with expert sarcoma knowledge and resources, including pathology, radiology, radiotherapy, surgery, medical, and pediatric oncology. However, because sarcomas are uncommon cancers, most physicians have little experience with diagnosis or treatment, and so the MDT is most effective when it is situated within a high-volume or dedicated reference center in a timely manner. This is illustrated by the above mentioned French NetSarc network [5]. Global implementation of MDTs varies considerably as a result of many countries lacking sufficient resources to implement such an approach. MDT management of sarcomas has also proven a greater challenge in smaller or less densely populated countries, such as Finland, where lower case volumes and geography hamper the practical implementation of dedicated specialist centers and MDTs.

Based on these factors, a common or specialized national or even international MDT approach could benefit patients, especially for those with rarer sarcoma subtypes. In Scotland, the operation of a Scottish Sarcoma Network allows patients to be referred to a National Sarcoma MDT, which connects five centers with sarcoma expertise across the country.

The COVID-19 pandemic has augmented the availability of virtual MDT models, such as that used by the Italian Rare Cancer Network (Rete Nazionale Tumori Rari) or the Transatlantic Australasian Retroperitoneal Sarcoma Working Group (TARPSWG). Together with advances in digital pathology, this approach could also open the door to further international MDTs for complex or rare cases.

Ensuring the appropriate medical expertise for each individual patient is a particular challenge in sarcomas. Focusing on surgery alone, there are many sarcoma cases where the surgery is relatively straightforward. Appropriate imaging acts as a guide to decisions about surgical margins, limb preservation, reconstructive surgery, etc. However, there are sarcoma patient populations where additional expertise is required. For example, head and neck sarcomas require a surgeon with appropriate experience in dealing with these anatomical structures; some pediatric limb tumors need additional orthopedic expertise; pelvic and spinal tumors present challenges; and within the retroperitoneal and abdominal spaces, it is now recognized that surgery by a sarcoma specialist with sufficient experience offers patients the best chance of cure [6].

The CTOS & SPAEN roundtable discussion focused on the following challenges:The need for a stronger body of evidence to demonstrate the status and benefits of MDT management of sarcomas. What different systems are operating globally? What impact does this have on prognosis? Performance data may be difficult to obtain but the structures operating within countries could be better understood and the examples of best practice could be showcased.The need to facilitate cross border collaborative healthcare with MDTs to support better patient care and also learning for healthcare professionals.

#### 2.2.2. Awareness/Education Amongst Healthcare Professionals

Sarcomas are often overlooked due to their rarity, or due to the young age of the patient. Reported experiences highlight a lack of knowledge of sarcomas among healthcare professionals and workers at the patient’s initial point of contact with a healthcare system. Most primary healthcare professionals have little or no experience in diagnosing sarcomas, and this lack of understanding often leads to symptoms being dismissed and patients being misdiagnosed. Primary healthcare professionals may also not know the specialist in their network. Primary care remains, in most instances, the first port of call for patients prior to their diagnosis. In the UK, survey data shows that 83% of sarcoma patients initially went to their General Practitioner when experiencing symptoms, but over a quarter were given incorrect advice [7]. Looking beyond primary care, other healthcare professional specialties with a role in sarcoma diagnosis include radiologists, sonographers, and physiotherapists, all of whom may encounter a case of sarcoma but not recognize the signs due to a lack of awareness.

The CTOS & SPAEN roundtable discussion focused around a single core challenge:

The need to define goals and focus efforts to raise awareness of sarcoma. Education for primary care professionals was highlighted as a priority. This global challenge requires a global, collaborative effort in order to achieve a meaningful impact.

### 2.3. Working on Cross-Border Challenges for Sarcoma Patients

#### 2.3.1. Background of Global Cancer Realities

Worldwide cancer cases will double to 37 million new patients per year by 2040. While access to and quality of global health care (including cancer and rare cancers, such as sarcomas) is already very unequally distributed between countries and healthcare systems, “modern oncology” is found almost exclusively in developed, high-income countries with a high standard of living. Developing, lower-income countries often lag behind. There is often a lack of national resources in finance, personnel, knowledge, structures, technologies, etc. The World Health Organisation (WHO) has already documented that cancer patients in poorer countries and poorer patients in rich countries have a lower chance of survival. Moreover, this global inequality of healthcare has just been underscored again by the COVID-19 pandemic.

#### 2.3.2. Background of Sarcoma Realities

If there is already a major imbalance in general health care and cancer, it is even more pronounced in rare cancers such as sarcomas. Even within Europe, rare cancers (including sarcomas) have significantly poorer survival rates compared with more common cancers (48.5% 5-year overall survival rate for rare cancers vs. 63.4% for common cancers) [8]. Even in Europe there are not many countries with designated sarcoma centers; there are even less “national networks of (certified) sarcoma centers”. Organizations like SPAEN receive emails on a daily basis from patients calling for help or looking for a 2nd opinion in “western” sarcoma centers. The critical questions are (1) what do patients do with their 2nd opinion when experts, therapies, or studies are not available in their home countries? And, more broadly, (2) what can patients without access to the correct diagnosis or treatment in their country do?

#### 2.3.3. Background for Future Solutions

The long-term solution for cross-border healthcare cannot be “medical tourism” for those patients who may be able to afford it. Rather: Solutions should be applicable on a regional/local basis; however, they need strong international cooperation and collaboration;A long-term solution could be the establishment of at least one sarcoma center (or sarcoma core team) per country or for neighboring countries. This would require knowledge transfer from experienced sarcoma centers to newcomers;For this type of cooperation across borders, digital health solutions will be crucial, such as solutions for diagnosis (pathology, radiology), remote surgery, data exchange, virtual tumor boards, or translation of medical reports;Existing initiatives such as EURACAN (European Reference Network (ERN) for rare adult solid cancers) in Europe for the exchange and discussion of patient cases or international financed projects such as SELNET (Sarcoma European & Latin American Network) for the exchange/transfer of experience and knowledge will become important;We also need better solutions for cross-border research such as digital (virtual) studies and registers;At a European level, investment should be placed in the development of training and educational activities for healthcare professionals and medical students in the usage of already existing tools that facilitate cooperation and knowledge-sharing within the medical community in a cross-border context, such as the Clinical Patient Management System (CPMS) [9].

However, the most limiting factor is the availability of resources. The solution cannot be to move more and more capacities to the West without reimbursement. Moreover, sarcoma centers are reaching their limits in capacity and finances. Solutions are needed that are adequately funded, that encourage initiatives or the commitment of individuals, support cooperation, and enable the bridging of borders and languages. 

### 2.4. Driving Research in Sarcomas

It is noteworthy that since the late 1980s, patients should be more involved in the development of cancer research [10,11]. The specific insights that are most valuable, as well as how to best gain these insights, continue to evolve [12,13,14,15]. Partnering with laboratory and clinical researchers to appropriately provide the patient voice is of keen interest to the sarcoma advocacy community. There are several important areas for driving forward sarcoma research together. Addressing these areas can lead to a joint approach for overcoming obstacles and barriers. Working together, advocates and researchers can strive to improve outcomes for people with sarcoma. The contributions of patients in the interpretation of research with the national health agency is another important aspect of research. The following is a preliminary list of challenges discussed during the meeting:

1.There Are No Clinical Trials in Large Parts of the World

Given the different approaches to cancer care and research across the world, access to clinical trials is not readily available across global borders. When standard approaches for the management of sarcoma are no longer effective, patients and families will seek new approaches that often are only available through a clinical trial. Therefore, identifying barriers and approaches to overcome obstacles is paramount to ensure access to the best possible treatment for people with sarcoma worldwide. The CTOS & SPAEN roundtable identified and discussed the following obstacles to sarcoma clinical trial global access: (1) Conduct of rigorous, safe clinical research requires infrastructure to ensure proper conduct and protocol compliance and safety of patients. This is not available worldwide. (2) The conduct of a single trial done across global sites can be fiscally prohibitive. (3) Access to study drugs may be limited due to regulatory requirements. 

2.The Incorporation of the Patient Voice in Planning Research/Clinical Trials

Collectively, researchers and patient advocates support working closely together to ensure that patient perspectives are incorporated into research planning. Sarcoma advocates acknowledge that an understanding of the scientific method is imperative to providing useful information at the appropriate inflection points. Advocacy groups are committed to expanding educational initiatives to prepare patient advocates to be research advocates [16]. There are many well-regarded models and resources available for training. One example of effective advocate engagement in cancer research planning is the US National Cancer Institute (NCI) Research Advocacy Council [17]. They have outlined key roles for research advocate engagement including: (1) Advise (share patient experience, patient derived real-world evidence), (2) Design (feasibility of the plan, assess for obstacles to participation), (3) Review (consent forms, patient facing information,) and (4) Disseminate (information, raise awareness, share results).

While time did not permit discussion at this meeting of all the potential obstacles for collaboratively driving research forward, the following additional items were noted and will be considered as the group continues to discuss opportunities to work together on sarcoma research planning:
Industry leading research with a strong focus on drugs and not enough research on other techniques or strategies.

○New drugs owned and directed by the pharmaceutical and biotech industry.
▪Researchers and advocates can provide valuable information to the industry on sarcoma and encourage seeking approval for this rare disease.▪While factoring in industry business needs, partner to identify mutually beneficial goals for all.
○Funding for research for multi-modality treatment is limited.
▪Generally, multi-modality research is driven by academic cooperative groups as interest across the industry has been more focused on drug development. Advocates can leverage their voices to lobby for more support.

2.Drug Development
○Too much research for Me-too drugs?
▪Once a target has proved beneficial in a disease, other similar compounds are studied with the aim of seeking approval.▪What are the downsides of having several drugs with similar mechanisms/targets for the same subtype?▪Given there are limited funding resources, does this divert funding from other subtypes?
○Looking for the lowest bar in drug development results in minimally beneficial drugs that are costly.
▪It can make business sense to have low bars to meet business goals of approval so a drug can be marketed. However, what is the measure of benefit and who defines this? Is this an opportunity for incorporating patient perspectives?▪Cost can be in dollars and side effects. Both need to be explored along with approval.
○Drug repurposing
▪In theory, there should be a shortened timeline to making a “repurposed” drug available if effective. This assumes that the dose and side effect profile will be the same in the patient population (may not be so). What is the potential for repurposing?▪Identifying sound rationale for the pursuit of a particular drug for sarcomas may be challenging.

3.Patient Reported Outcomes (PRO) in rare diseases—the “right” PRO for evaluation of new therapies
○This field is evolving with several sarcoma groups developing subtype specific PRO tools. The European Organisation for Research and Treatment of Cancer (EORTC) and the Patient Reported Outcomes Measurement Information System (PROMIS) have validated tools that can be adapted.○Is there an opportunity to have treatments approved with PROs as a primary endpoint?


While there are many challenges to the conduct of research in rare cancers such as sarcomas, these present researchers and advocates have an excellent opportunity for collaboration, working together to overcome the obstacles and achieve better outcomes for people with sarcoma.

## 3. Priority List of Projects to Be Worked on Together over the Next 5 Years

The CTOS & SPAEN roundtable meeting discussion focused on the possible issues that might help improve the sarcoma patient’s treatment pathways and outcomes, both nationally and internationally, all of which would need to be underpinned by evidence from research:

### 3.1. Minimum Requirements for Sarcoma Care and Sarcoma Specialist Centers and Empower Patients to Find Them

There is a need to define and standardize a “sarcoma specialist center or network”. The definition is currently varying broadly with very different levels of expertise and experience (if any). This calls for a global charter of minimum requirements that a sarcoma center (or a sarcoma specialist) should meet. 

When it comes to standardizing experience and expertise, a ‘graded’ approach may be applicable. However, the criteria must be standardized internationally and with respect to national variations in healthcare systems so that they can be adopted without unnecessary barriers being applied. The criteria should be varied and define centers as reference centers, e.g., number of patients seen, outcome of patients, experience of the team, research record of the team, etc. The entire body of criteria will need coordination and presentation as a single proposal. 

However, poorly resourced countries do not and will not have specialists, and maybe not even single specialist doctors. There’s a high need for linking non-experts to experts in countries with sarcoma expertise. 

Use of digital resources should be used to empower patients to find a specialist team (or a specialist doctor, where no expert center is available), to consult using technology (telehealth methods) where appropriate, and to know how to act on the information and advice given. This is the logical adjunct and will require resources. The concept of an ‘atlas’—a global database of “Sarcoma Treatment Expertise”—has been discussed.

### 3.2. Digitally Enabled Multidisciplinary Networks and Stronger Body of Evidence for Benefits of MDT Sarcoma Management

Digitally enabled multidisciplinary networks would most likely be regional/national but might include international members on a case-by-case discussion or to cover extremely rare conditions. Such networks may operate at a primary care level to accelerate diagnosis, at a secondary care level to facilitate appropriate treatment by non-specialists, as well as at a tertiary care level to support decision-making with rare tumors and more complex presentations. There is a case for international networking that supports pathology. There are such “virtual MDT” networks already in existence in several countries. Structured reviews of each network together with an overview of the evidence they present are needed.

### 3.3. Establish Sarcoma Expertise, Facilitate Access to Established Sarcoma Centers and Cross-Border Healthcare

In small or poorly resourced countries, there will not be an established sarcoma management system in place—or most likely not even sarcoma specialists. There is a high need internationally to educate and establish links to experts in order to build up knowledge and capacity and to facilitate access of teams in centers not equipped with a sarcoma team to a sarcoma reference center, preferably in their country. If this is impossible, links to one of the EURACAN centers or equivalents elsewhere in the world should be established. These teams should be integrated in training programs. In addition, cross-border collaborative healthcare should be established or fostered where needed for selected patients, complex surgeries, and rare technologies (e.g., proton beam, interventional radiology, clinical trials, etc.). However, what is most crucial is to enable in-country care for sarcoma patients. 

### 3.4. Develop a Global, Collaborative Awareness Effort in Order to Achieve a Meaningful Impact

Awareness is a huge issue everywhere. It is agreed that it is impractical to inform the public as a whole. The particular groups where most value can be achieved are probably in primary care, medical education, and community healthcare (e.g., physiotherapy). This may be best achieved through linking with other rare cancers rather than pursuing it specifically in sarcomas. Various awareness projects have been tested in different countries (e.g., golf ball in the UK and Poland). There is a need to review these projects to document what worked well, what did not work, and the lessons learned in order to shape future endeavors. 

### 3.5. Global Access to Sarcoma Clinical Trials and Incorporating the Voice of Sarcoma Patients (Advocates)

The potential approaches identified for overcoming some of the above outlined obstacles to sarcoma clinical trial global access include researchers designing jointly clinical trial protocols with the intent of conducting the trial independently in various parts of the world. Conducting parallel trials is one approach that will allow each site/country to align with local regulatory requirements while prospectively planning a pooled analysis of the data. While this may not overcome all obstacles for all sites, this should aid in expanding global access to clinical trials. Sarcoma advocates can leverage the strength of their voices in support of sarcoma research initiatives by advocating to pharmaceutical companies to provide access to drugs for clinical research globally. Armed with the knowledge and expertise of the sarcoma research community, sarcoma advocates can speak firsthand to the meaning and importance of access to new treatments for rare cancers such as sarcoma. In summary, it should be the objective to bring patients to clinical trials and clinical trials to patients, regardless of where they live. 

### 3.6. Further Ideas and Potential Deliverables over the Next 5 Years

○The continuation of expert–patient advocate “consensus roundtables” for sarcoma subtypes○A position paper on future optimal research and care for sarcoma patients○Foster the establishment of sarcoma PAG in more countries worldwide

## 4. Conclusions

This meeting offers an excellent starting point for strengthening the collaboration between patient advocates and experts with the collective goal of improving outcomes for people with sarcomas.

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
