# Peer review of "Global Patient Involvement in Sarcoma Care—A Collaborative Initiative of the Connective Tissue Oncology Society (CTOS) & Sarcoma Patients EuroNet (SPAEN)"

_cancers, 2022, doi:10.3390/cancers14040854_

Round 1

Reviewer 1 Report

This position paper summarizes an important initiative, bringing together patient advocates and caregivers will be key to future successes. The authors should be congratulated for taking the initiative. I have a few comments:

Page 1, last line, should read "The objective of their work with experts...", as not all authors are part of of SPAEN.

The statement be Roger Wilson on page 2 should be shortened, this is not an elogy but a scientific paper after all.

In-text citation format is not correct (--> no upper-case numbers in brackets).

Page 8 first paragraph, imperative voice should not be used. "crucial" should be used instead of the adverbial form.

Author Response

  • The sentence on page 1 "The objective of their work with experts ..." has been corrected accordingly.
  • The statement by Roger Wilson on page 2 has already been shortened to fit into the manuscript and is of utmost importance. Therefore, we would like to retain this as it is.
  • The citation format has been corrected in the text as recommended.
  • On page 8 the imperative forms have been removed and the form "crucial" was used as suggested.

Reviewer 2 Report

The manuscript described the perspective of sarcoma care at CTOS and SPAEN. The topic is important and critical for the field of sarcoma.

some minor comments

1) The paragraph of early and accurate diagnosis

 The authors emphasize the accurate pathological diagnosis. But, I think the accurate biopsy is also important.

2) MDT conference

All should agree with the importance of MDT. How about the timing of the MDT conference? Ideally, when the physicians or/and patients demand, MDT should be discussed as soon as possible. But, even if the web MDT conference can be done, it should be difficult for all related resources to participate in the conference in a short duration. 

3) Awareness/education among healthcare professionals

I also think healthcare professionals may not know the specialist in their network. For example, if the GP finds a lump at the neck lesion, which is appropriate, general surgeon, orthopedic surgeon, or head and neck surgeons?  

Author Response

  • Page 2: The importance of an accurate biopsy has been added as recommended.
  • Page 3: A statement about timing of the MDT conference has also been added as suggested.
  • Page 4: The statement that "Primary healthcare professionals may not know the specialist in their network." has been added to the paragraph as recommended.

Reviewer 3 Report

This perspective manuscript by Kasper et al. hass been well organized and written. The main findings and possible future steps in the meeting have been included and highlighted, which offer an idea and resource for readers with diversified background and knowledge.

I have only one minor concern/question about how the readers such as people from patient advocacy groups understand the R classification system in Page 2.  I wonder whether the sentence can be edited to help reader understand, e.g. “the rate of R0 curative resection is 50% lower compared to patients operated within a NetSarc cen-ter, while the rate of R2 resections with macroscopic residual tumor is more than doubled”.

Author Response

Page 2: The recommended additions explaining the R classification system have been added accordingly.